# Structural Role of Plasma Membrane Sterols in Osmotic Stress Tolerance of Yeast *Saccharomyces cerevisiae*

**DOI:** 10.3390/membranes12121278

**Published:** 2022-12-17

**Authors:** Svyatoslav S. Sokolov, Marina M. Popova, Peter Pohl, Andreas Horner, Sergey A. Akimov, Natalia A. Kireeva, Dmitry A. Knorre, Oleg V. Batishchev, Fedor F. Severin

**Affiliations:** 1Belozersky Institute of Physico-Chemical Biology, Lomonosov Moscow State University, 1-40 Leninskie Gory, 119991 Moscow, Russia; 2Frumkin Institute of Physical Chemistry and Electrochemistry, Russian Academy of Sciences, 31/4 Leninskiyprospekt, 119071 Moscow, Russia; 3Institute of Biophysics, Johannes Kepler University Linz, Gruberstraße 40, 4020 Linz, Austria

**Keywords:** sterol, hyperosmotic stress, hypoosmotic stress, yeast, giant unilamellar vesicle, large unilamellar vesicle, light scattering

## Abstract

Yeast *S. cerevisiae* has been shown to suppress a sterol biosynthesis as a response to hyperosmotic stress. In the case of sodium stress, the failure to suppress biosynthesis leads to an increase in cytosolic sodium. The major yeast sterol, ergosterol, is known to regulate functioning of plasma membrane proteins. Therefore, it has been suggested that the suppression of its biosynthesis is needed to adjust the activity of the plasma membrane sodium pumps and channels. However, as the sterol concentration is in the range of thirty to forty percent of total plasma membrane lipids, it is believed that its primary biological role is not regulatory but structural. Here we studied how lowering the sterol content affects the response of a lipid bilayer to an osmotic stress. In accordance with previous observations, we found that a decrease of the sterol fraction increases a water permeability of the liposomal membranes. Yet, we also found that sterol-free giant unilamellar vesicles reduced their volume during transient application of the hyperosmotic stress to a greater extent than the sterol-rich ones. Furthermore, our data suggest that lowering the sterol content in yeast cells allows the shrinkage to prevent the osmotic pressure-induced plasma membrane rupture. We also found that mutant yeast cells with the elevated level of sterol accumulated propidium iodide when exposed to mild hyperosmotic conditions followed by hypoosmotic stress. It is likely that the decrease in a plasma membrane sterol content stimulates a drop in cell volume under hyperosmotic stress, which is beneficial in the case of a subsequent hypo-osmotic one.

## 1. Introduction

In the yeast *S. cerevisiae*, hyperosmotic stress causes a decrease in intracellular ergosterol concentration [1]. This suppression is mediated by the key yeast high osmolarity sensor kinase, Hog1, which activates the transcription of Mot3. The latter protein is a transcriptional suppressor of a set of genes involved in the ergosterol import and the biosynthesis (as reviewed in [2]). In agreement with that, it has been shown that the suppression is essential for yeast cell survival [1]. In addition, various mutations affecting the ergosterol biosynthetic pathway render yeast cells sensitive to hyperosmotic stress [3,4]. Generally, it is believed that the physiological role of sterols in an osmoadaptation is the adjustment of the activities of plasma membrane pumps and channels (reviewed in [2]). However, it has also been suggested that sterol content may affect mechanical properties of the plasma membrane (PM) and thus influences the survival upon a change in an osmotic pressure [4]. Similarly, a gadolinium-mediated modulation of a membrane fluidity has been shown to be involved in a formation of water pores in A549 cells after hypotonic cell swelling [5]. The latter line of reasoning seems more attractive than the former one, since the concentration of sterol can reach fifty percent of the PM total lipid content, implicating that the primary role of sterols in the PM is most likely structural. Indeed, one can speculate that it should take less time and resources to adjust the activity of PM pumps and channels by phosphorylation rather than by transcriptional changes in the ergosterol biosynthesis. Following this logic, in the current work we attempted to estimate physicochemical changes of lipid bilayers caused by a decrease in a sterol content and to match these changes with potential benefits for cells upon the occurrence of osmotic stress. In fact, these benefits are not so obvious. It is known that the concentration of sterols in the PM is much higher than in membranous organelles inside the cell. It has been shown that sterols provide lipid bilayers with additional rigidity, resistance to rupturing, and also decrease the permeability to hydrophilic molecules [6,7,8,9,10]. All of these features are believed to contribute to the barrier function of the PM [11].

Assuming that lowering the PM sterol concentration plays a structural role upon hyperosmotic stress, one might speculate that decreasing the rigidity with simultaneous increase of the water permeability of the membrane might be beneficial, allowing the “deflation” of the PM upon hyperosmotic stress (Figure 1). The volume decrease causing the membrane folding upon the occurrence of hyperosmotic stress might equilibrate the osmotic pressure. An enhanced rigidity due to high sterol concentration is likely to oppose the folding. In addition, lower permeability of the sterol-rich membrane to water is likely to decelerate the adjustment (decrease) of the cell volume in hyperosmotic conditions. Thus, sterol-rich membranes seem to be more prone to rupturing (i.e., a pore formation) upon the occurrence of hyperosmotic stress followed by the hypoosmotic one (Figure 1). Importantly, it is known that the free energy of an intact lipid bilayer is lower than that of the pore-containing one [12]. Thus, pores forming in the sterol-rich membrane upon the occurrence of hyperosmotic stress are expected to be able to self-repair (Figure 1). At the same time, sterol-rich membranes are characterized by the approximately an order of magnitude higher modulus of lateral stretching [13] meaning that the membrane rupture occurs at smaller relative changes of the enclosed volume. In the case of sterol-free membranes and a vesicle swelling, induced by hypotonic solution, membrane rupture occurs already above the membrane area increase of ~3% [14]. Yet, such numbers cannot be directly translated to cells as the plasticity of their membranes is increased by membrane reserves [15].

In the present study we tried to clarify the influence of the sterol content on the membrane integrity with respect to transient hyperosmotic gradients. We used large and giant unilamellar vesicles (LUVs and GUVs, respectively) as model membrane systems, as well as yeast cells. We utilized fluorescent microscopy on GUVs, which are known as a good model of the lipid matrix of the cell PM, to illustrate possible changes of the GUV structure upon propagation of the hyperosmotic front. Additionally, we performed stopped-flow experiments in hyperosmotic conditions on small unilamellar vesicles, the membrane of which either contained or not contained cholesterol. Finally, we studied the hyper- and hypoosmotic stress on wild-type yeast, as well as on mutants having elevated or lowered levels of ergosterol in the plasma membrane. Our data indicate that sterols increase rigidity of the LUVs, GUVs and the plasma membrane. This additional rigidity prevents swelling upon hyperosmotic stress, which, in turn, promotes the membrane rupturing upon the following kind of hypoosmotic stress.

## 2. Materials and Methods

### 2.1. Chemicals

*E. Coli* polar lipid extract and cholesterol were purchased from Avanti Polar Lipids (Alabaster, AL, USA). Chloroform (>99.0%), sodium chloride (NaCl), sucrose, sorbitol, and HEPES were purchased from Sigma (St. Louis, MO, USA). All chemicals were used without further purification. d-glucose (Roquette-361,103-0.5) was purchased from Helicon. Bacto Agar (0207/0-PW-L.500) and peptone (HYP-A.5000) were obtained from BioSpringer.

### 2.2. Large Unilamellar Vesicle Preparation

Large unilamellar vesicles (LUV) were prepared as described in [14]. In brief, a lipid solution containing the *E. Coli* polar lipid extract (PLE) or PLE with 23 mol.% of cholesterol dissolved in chloroform was rotary evaporated on the bottom of a glass flask and kept under vacuum for one hour. The resulting dried lipid film was resuspended in the working buffer solution (10 mM NaCl, 140 mM KCl, 50 mM HEPES, pH 7.48). After vigorous vortexing the solution was extruded 21 times through two polycarbonate membranes with a pore diameter of 100 nm (Avestin, ON, Canada) to give unilamellar liposomes with an average diameter of 120 nm–150 nm [14,16].

### 2.3. Stopped Flow Experiments

Equal volumes of vesicle suspension in the buffer solution and hyperosmotic solution (buffer solution with 150 mM sucrose or 150 mM sorbitol added) were mixed at 5 °C. The scattered light intensity *I*(*t*) at 90° was measured in a stopped-flow apparatus (SFM-300, Bio-Logic, Claix, France) with a dead time of 2.6 ms at a wavelength of 546 nm.

As a result of the osmotic stress, water left the vesicles by a passive diffusion through the lipid membrane. We denote the membrane permeability for water as *P_f_*. The change in the vesicle volume over time can be written as [17]:(1)$$\frac{dV\left(t\right)}{dt}=AP_{f}V_{w}\left(\frac{V_{0}}{V\left(t\right)}c_{0}^{i}-\left(c_{0}^{i}+c_{S}\right)\right),$$
where *V_w_*, *V*_0_, *A*, *c*_0_*^i^* and *c_s_* are the molar volume of water, vesicle volume at time zero, area of the vesicle membrane, the initial osmolarity of the solution inside the vesicles, and the incremental osmolarity in the external solution due to sucrose or sorbitol addition, respectively. The analytical solution of Equation (1) has the form:(2)$$V\left(t\right)=V_{0}\frac{c_{0}^{i}}{c_{0}^{i}+c_{S}}\left\{1+L\left(\frac{c_{S}}{c_{i}^{0}}\exp\left(\frac{c_{S}}{c_{0}^{i}}-\frac{AP_{f}V_{w}{\left(c_{0}^{i}+c_{S}\right)}^{2}}{V_{0}c_{0}^{i}}t\right)\right)\right\},$$
where *L* is the Lambert function, defined as: $L\left(x\right)e^{L\left(x\right)}=x$. The explicit dependence of *I*(*t*) on *V*(*t*) is given by the Rayleigh-Gans-Debye approximation [18], and is rather cumbersome. However, it has been shown in [19] that the exact dependence can be quite accurately approximated by its Taylor series up to the quadratic term:(3)$$I\left(t\right)=a+bV\left(t\right)+cV{\left(t\right)}^{2}$$
with constant (time-independent) coefficients *a*, *b*, *d*, which can be obtained analytically. Thus, the only unknown parameter *P_f_* can be found from the fit of the experimentally determined dependence *I*(*t*) by Equation (3). However, as demonstrated in [19], technically it is simpler to obtain all coefficients (*a*, *b*, *d*, *P_f_*) directly from the fit. We performed the fit using the gradient descent method, and thus determined *P_f_* for membranes of different compositions (PLE or PLE + cholesterol) treated by different osmolytes (sucrose or sorbitol).

### 2.4. GUV Experiments

GUVs were obtained by an electroformation technique on conductive ITO-coated glass slides [20]. In brief, 5 μL of a lipid solution in chloroform (compositions: (1) 99.9 mol.% of DOPC + 0.1 mol.% of Rhod-PE; (2) 69.9 mol.% of DOPC + 30 mol.% of ergosterol/cholesterol + 0.1 mol.% of Rhod-PE) were applied over each conductive side of the slides with an area of 4 cm^2^, and dried under a stream of argon for 2 min. Then, the slides were separated with a 1 mm PDMS spacer and the cell was filled with the buffer (220 mM sucrose, 2 mM NaCl, and 1 mM HEPES, pH 7). A sinusoidal voltage with an amplitude of 2 V and a frequency of 11 Hz was applied to the ITO-coated glass slides at 45 °C for 3 h.

GUVs were observed using a Nikon Ti-E fluorescent microscope with a 60× objective. For experiments, 10 μL of the GUV suspension were placed in 100 μL of the isosmotic buffer (100 mM NaCl, 5 mM MES, pH 7) at a cover glass pretreated by bovine serum albumin to avoid the GUV disruption [21]. To study the effect of hyperosmotic conditions, a hyperosmotic solution (2 M NaCl, 5 mM HEPES, pH 7) was added to the GUV using a microinjection glass pipette. Recorded movies were further processed using ImageJ software.

The effect of hyperosmotic conditions on the GUV was analyzed by measuring the relative change in the average vesicle diameter before, during, and after application of the hyperosmotic buffer. Hyperosmotic buffer was added using a micropipette with a diameter of 1 μm for 3 s. In such conditions, the released hyperosmotic solution represents a cloud that passes in the vicinity of the GUV membrane, and locally raises the salt concentration. About 20 GUVs of the each lipid composition were chosen for measurements, the average diameter of the each vesicle was measured by 10 snapshots, with two diameters in perpendicular planes for the each GUV on the snapshot.

The relative change in the GUV size was calculated according to the equation:(4)$$A=\frac{d_{0}-d_{1}}{d_{0}}\times 100\%,$$
where *d*_0_ and *d*_1_ are average diameters before and during (after) NaCl addition, respectively.

### 2.5. Yeast Strains and Growth Conditions

We used the standard rich yeast medium yeast peptone dextrose (YPD) described by Sherman [22]. Yeast cells were grown in liquid YPD overnight in 50 mL sterile tubes up to the exponential growth phase. Next, we equalized the yeast cell concentration of the different mutants by diluting the cultures with fresh YPD to ensure a final OD550 of 0.05 that corresponded to a cell density of 10^6^ cells/mL. The cells were then transferred into 96-well microplates (Eppendorf, 0030730011; Hamburg, Germany) at aliquots of 100 μL and incubated in a SpectroStar Nano (BMG Labtech GmbH; Hamburg, Germany) microplate reader. The optical density was measured every 5 min for the entire duration of the experiment (20 h), with the temperature set at 30 °C. The cells were shaken for a minute before the OD measurements (500 rpm). Table 1 summarizes the yeast strains used in this study.

### 2.6. Deletion of LAM2

*LAM2* deletion in the *UPC2-1* background was obtained by the homologous recombination of the PCR product (primers: LAM2-400 5′-CGTTAGTCCACCATAACCAA and LAM2+200 5′-AGTAATGCACCAGAAATGGA) with the heterologous selection marker HIS3. The templates used during PCR were DNA of the *Δ**lam2* strain. The disruption cassette integration and the deletion of *LAM2* gene were verified by the PCR with independently designed primers flanking the disruption cassette: (primers: LAM2-600 5′-CGTTTAATATCGTCAACGAC and LAM2+300 5′-CCAGATATAGATGCTATATG).

### 2.7. Growth Kinetics Analysis

The growth kinetics analyses were conducted in a semi-automatic manner using a custom R-script. μmax was calculated as the maximum detected slope of Log2 (OD) during the course of the experiment. Therefore, the μmax values are equal to 1/duplication time. The lag period was defined as the interval between the start of the experiment and the time point when the cells reached their maximal growth rate.

### 2.8. Propidium Iodide Uptake upon Hypoosmotic Stress

The overnight cell cultures of three strains (*W303*, *UPC2-1*, *Δ**lam1**Δ**lam2**Δ**lam3**Δ**lam4* (further referred to as *Δ**lam1234*)) were transferred to 1.5 mL centrifuge tubes, centrifuged at 4000 rpm (~1300 g) and resuspended in fresh YPD media until OD_550_ = 1 (approximately, 2 × 10^7^ cells in 1 mL). Then the NaCl solution was added until the concentration reached 330 mM in all samples. Subsequently, the samples were incubated for 5 min at room temperature. After that, the samples were centrifuged again, and the media was replaced with milli-Q water with propidium iodide (PI) (1 μL [0.1 μg/mL PI] per 100 mcl = 1 × 10^−3^ μg/mL). 10 min after adding propidiuim iodide, the samples were diluted and analyzed via flow cytometry in channel PC5.5-A(CytoFLEX).

## 3. Results

### 3.1. Lowering Sterol Content of Artificial Liposomes Increases Their Water Permeability

It has been reported that the water permeability of liposomal cholesterol-free membranes is higher than the permeability of cholesterol-containing ones [6]. As this finding is apparently in an agreement with our hypothesis, we attempted to confirm these data using sorbitol and sucrose as osmolytes.

In cases when sorbitol is used, the time dependent scattering intensities of cholesterol free vesicles are non-monotonous (Figure 2a). It seems that in the absence of cholesterol sorbitol is able to penetrate the membranes [14], whereas on the timescale of the experiment sucrose does not penetrate the membranes (Figure 2b). In the case of the non-monotonous scattering data, we determined the membrane permeability with respect to water fitting the rising left part of the scattering curve *I*(*t*), from *t*~0 till the maximum of the scattering intensity [14]. The membrane permeability with respect to water determined from the fits is *P_f_* = 12.7 µm/s both for sucrose and sorbitol used as an osmolyte. Addition of cholesterol (Figure 2c,d) led to an approximately two-fold decrease in the permeability (12.7 µm/s vs. 5.7 µm/s). Note, that from Equation (2), the characteristic time of the LUV shrinkage is of the order of 1 s for *P_f_* = 5.7 µm/s, and about 0.5 s for *P_f_* = 12.7 µm/s. However, if we substitute into Equation (2) the parameters typical for GUVs, the corresponding characteristic times would be about 200 s and 100 s, respectively, under the same osmotic conditions. In real experiments on the water leakage from GUVs these time values could be even larger due to unstirred layers adjacent to the membrane, the width of which can reach hundreds of micrometers [26].

Thus, sterol addition decreases the water permeability of artificial lipid bilayers. However, does that explain why the cells reduce the sterol content of their PM upon hyperosmotic stress? Our next step to answer this question and to test our hypothesis (Figure 1) was to study the response of giant unilamellar vesicles) to the addition of high salt.

### 3.2. Sterol Prevents Shrinkage of Giant Unilamellar Vesicles upon Transient Hyperosmotic Stress

We have challenged sterol-free and sterol-rich GUVs with the transient hyperosmotic stress applied by a finite-time lasting flux of hyperosmotic buffer from a vicinal micropipette. Changes of the GUV volume were monitored throughout the entire experiment. According to our hypothesis, we expected a more pronounced shrinkage of the sterol-free GUVs due to the smaller elastic rigidity and the higher water permeability with respect to cholesterol-free membranes.

In the case of pure DOPC GUVs, a vesicle shrinkage resulted in the diameter decrease of (12 ± 5)% in approximately 15 s after the start of the application of the hyperosmotic solution, with the formation of spherical daughter vesicles inside the GUVs. After the hyperosmotic front passed, the vesicle size was restored (30 s from the beginning of the experiment) (Figure 3A). For GUVs with ergosterol or cholesterol, only minor relative changes in the GUV size occurred at this point of time (15 s after the start of the application of the hyperosmotic solution), despite the formation of “beads on a wire” such as protrusions inside the vesicle (Figure 3B,C). At the same time, formation of the protrusions could add to the volume stability of cholesterol-rich GUVs [27]. As the membrane area stored in such protrusions is negligible compared to the surface area of GUVs, formation of such protrusions does not contradict with minor changes in GUV diameters. The only observable difference between ergosterol and cholesterol in our in vitro experiments concerned the time of appearance of protrusions in hyperosmotic conditions. In the case of cholesterol, they formed twice as late (see Figure 3). The results are summarized in Table 2. Substitution of NaCl with KCl in the hyperosmotic solution did not result in any changes.

Therefore, in agreement with our model (Figure 1), sterols appear to provide GUV membranes with additional rigidity and smaller water permeability thus preventing the volume decrease during and right after the transient hyperosmotic challenges, in contrast to pure DOPC vesicles.

### 3.3. Yeast Mutants in Sterol Biosynthesis/Trafficking Are More Sensitive to Sodium Chloride than to Sorbitol

The results of the experiments on LUVs and GUVs are, apparently, in agreement with our model (Figure 1). Next, we used yeast cells to verify the model. According to our scheme, membranes with high sterol content are envisioned to undergo transient pore formation upon hyperosmotic stress until the osmotic pressure is released. Thereby, equilibration of the osmotic pressure across the cell membrane can be achieved via the influx of salt or other osmolytes from the external medium or water efflux from the cell interior. Thus, one can expect that the chemical nature of the osmolyte, which may flood the cell in an uncontrolled manner, will affect the survival and/or growth rate of the cells with the high sterol PM. To test this hypothesis, we used three osmolytes of a different chemical nature: sodium chloride, potassium chloride, and sorbitol. Whereas potassium, chloride and sorbitol are major cytosolic ions, sodium is less physiological. Therefore, we expected that while the cells with the normal PM sterol content would be equally sensitive to all of these osmolytes, the cells with elevated PM sterol would be more sensitive to NaCl. Hence, we estimated growth rates of a set of mutants in the media containing 0.6 M NaCl, 0.6 M KCl or 1.2 M sorbitol. These concentrations were chosen to (i) provide equal osmotic pressures and (ii) to cause a moderate delay in the growth rates. The results are summarized in Figure 4 with exemplary growth curves being visualized in Appendix A (https://www.mdpi.com/article/10.3390/membranes12121278/s1). In accordance with our prediction, the growth rate in the wild type control cells of *BY4742* genetic background was similarly delayed by all three osmolytes. Interestingly, sorbitol caused stronger growth delays in the wild type control of *W303* genetic background and in the hog1 deletion strain. Hog1p is the key transcription factor responsible for the cellular defense against the high osmolarity stress (reviewed in [28,29,30]). Possibly, this is due to the fact that, unlike NaCl or KCl, sorbitol can be metabolized by cells and thus may cause some additional effects distinct from its action as an osmolyte [31]. To our knowledge, the influence of *hog1* deletion on sorbitol metabolism has not been studied yet. Possibly, the rate of sorbitol catabolism is affected by the deletion thus affecting the sensitivity. We have also included in our analysis the strains from our collection of mutants in ergosterol biosynthesis and/or transport. Out of 17 strains (Appendix A, https://www.mdpi.com/article/10.3390/membranes12121278/s1), three strains (*Δ**erg4**Δ**lam2*, *UPC2-1*, *UPC2-1**Δ**lam2*) have displayed differential sensitivities to the three osmolytes, as depicted in Figure 4. Whereas Erg4p catalyzes one of the final steps of ergosterol biosynthesis (reviewed in [32]), Lam2p is a transporter, which pumps the excess of PM sterol inside the cell. Upc2p, on the other hand, is the key transcription factor regulating the sterol biosynthesis with the mutation rendering the sterol biosynthesis hyper-active [23]. Thus, the PM sterol concentration of all three strains is likely to be elevated, which is consistent with our model (Figure 1). The sensitivities of the rest of the strains were not significantly different (Appendix A). In all three cases, the sensitivity to NaCl was significantly higher than to KCl or sorbitol (Figure 4). In other words, some of PM ergosterol disruptions caused a relative increase in the sensitivity to NaCl. None of such mutations caused a relative increase in the sensitivities neither to KCl nor to sorbitol. Therefore, these results are consistent with the hypothesis that alterations in PM sterol content cause osmolyte influx upon the occurrence of hyperosmotic stress.

### 3.4. Yeast Cells Deficient in the Reverse Transport of Sterol from the PM Accumulate Propidium Iodide upon Hypoosmotic Stress

Our hypothesis, illustrated by Figure 1, predicts that the PM of yeast cells with the high PM sterol content is likely to lose its integrity upon the hypoosmotic stress, which follows the hyperosmotic one. To test this, we subjected cells with normal or elevated PM sterol to a mild high salt stress (0.4 M NaCl), and then transferred them into a medium with regular ionic strength (0.05 M NaCl). We used two strains with elevated PM sterol. One of them, *UPC2-1*, is a mutant carrying a hyper-active allele of the key transcription factor, Upc2, responsible for ergosterol biosynthesis (reviewed in [2]). Another strain has been previously characterized by our group as carrying a quadruple deletion of the LAM 1-4 genes responsible for the reverse sterol flow from the PM thus displaying high PM sterol [23]. To monitor the loss of integrity we added propidium iodide (PI) to the latter solution. The results presented in Figure 5 show that, indeed, while the cells of the control strain (*W303*, Figure 5) did not accumulate PI, the strain lacking the PM reverse sterol transporters (*Δ**lam1**Δ**lam2**Δ**lam3**Δ**lam4* (lam 1234), Figure 5) displayed a significant proportion of PI-positive cells. Unexpectedly, the mutant *UPC2-1* cells did not accumulate PI (Figure 5), despite elevated PM ergosterol [23]. Possibly, the activity of the reverse transporters in this mutant, Lam proteins, provided local relaxation of the PM rigidity. In other words, despite the total elevated sterol content, even local low-sterol PM patches might provide sufficient flexibility to allow, as shown by Figure 1, the PM shrinkage upon the hyperosmotic stress, and in this way prevented the influx of the PI-containing external medium, when the osmotic pressure was normalized.

## 4. Discussion

The PM of eukaryotic cells typically exhibits a higher sterol content than the other membranous organelles. It is believed that sterols contribute to the barrier function of the PM, and, in particular, provide an additional resistance against rupturing [11]. As osmotic pressure exerts mechanical load at the PM, one would expect that a cell would increase, not decrease, the PM sterol content upon an osmotic stress, and this appears to be the case of plant cells. Membranes of salt-tolerant plants have been shown to be richer in sterols than the ones of salt-sensitive ones. Accordingly, it has been suggested that the ability to increase the total sterol content the under salt stress may allow salt tolerant species/genotypes to adapt to the high salt (reviewed in [33,34]). On the contrary, yeast cells decrease PM sterol upon osmotic stress [1]. It is possible that the difference between *S. cerevisiae* and plants is due to the fact that the outer surface of plant roots is sufficiently rigid to prevent the volume expansion upon hypoosmotic stress, which may follow the hyperosmotic one. Indeed, as illustrated in Figure 1, such a combination of stresses could lead to a massive rupturing of the PM.

However, the latter argument does not apply to a number of unicellular species of *Dunaliella*, a marine unicellular alga. These organisms are known to be adapted to high amplitude changes in salt concentrations. While their PM sterol content is high, it is known that they do not possess a rigid cell wall. It has been reported that they react to fluctuations in the osmotic pressure by rapid changes in the intracellular osmotic pressure via increasing the concentration of glycerol (reviewed in [35]). Indeed, it is generally known that a tolerance to a dehydration is, first of all, a synthesis of osmolytes. Interestingly, when the sterol synthesis is inhibited in *Dunalliela*, the glycerol biosynthesis is disrupted [36]. This points to an additional link between osmo-tolerance and sterols.

In the present study, we utilized model lipid membranes of LUVs and GUVs to comparatively study the effect of the osmotic stress on closed membrane envelopes in the presence and absence of cholesterol and ergosterol. In experiments with hyperosmotic shrinkage of LUVs, we demonstrated that sterols significantly reduce the water permeability of the membrane. At the same time, we did not detect a rupture of the membrane upon hyperosmotic stress in these experiments. Studies on LUVs correlated with the experiments on GUVs: sterols (both cholesterol and ergosterol) decreased the passive membrane permeability of water limiting the GUV volume change upon propagation of the hyperosmotic front. Sterol-increased rigidity of the membrane resulted in the formation of predominantly filamentous protrusions inside the giant vesicle, in contrast to spherical protrusions with bigger radii in sterol-free membranes. Therefore, we observed only minor changes in the GUV volume upon hyperosmotic challenges in the case of sterol-rich membranes. Significant changes in the GUV volume upon propagation of the hyperosmotic front may suggest a transient formation of pores in their membranes since they require a drastic drop in the volume/area ratio. Resealing of such pores was hampered in the case of an increased membrane viscosity, which is known to be related to the high sterol content of the PM [37]. Whereas sterol is always present in yeast membranes, some mutants have altered levels of PM sterol. Nevertheless, its molar fraction never reaches zero. Comparing the results obtained on yeast and model membranes, we thus assumed that the sterol influence on membrane properties was monotonic. In other words, the change in membrane properties upon an increase of the sterol level, say, from 0 to 20 mol.%, is qualitatively similar to that upon an increase of the sterol level from 30 to 50 mol.%. Thus, here we came to some interesting contradiction. Despite sterols decrease the membrane water permeability, the PM of cells with decreased sterol content is able to faster recover from the transient loss of the barrier function, comparing to sterol-free ones. Therefore, successive changes from hyperosmotic to hypoosmotic conditions might be much more harmful to the rigid and less fluid sterol-containing membranes. This is exactly the case we observed on yeasts (Figure 5). Noteworthy that timescales of recovering from even short (3 s) osmotic stress on GUVs cover tens of seconds (Figure 3). This might significantly influence a yeast metabolism, forcing cells to consume a lot of energy to maintain the PM integrity. Thus, even short transient changes in the osmotic gradient across the PM could have delayed effects in hours of yeast growth, as we observed in our experiments (Figure 4).

The yeast *S. cerevisiae* appeared to be unique in its strategy of adaptation to osmotic challenges. It decreases PM sterol content upon the occurrence of hyperosmotic stress. This increases the water efflux from the cell upon hyperosmotic stress, which, obviously, if taken separately, is detrimental for the cell’s physiology. At the same time, our data suggest that such efflux provides an insurance in the case of the hyperosmotic stress is being followed by the hypoosmotic one (as illustrated by Figure 1).

## Figures and Tables

**Figure 1 membranes-12-01278-f001:**
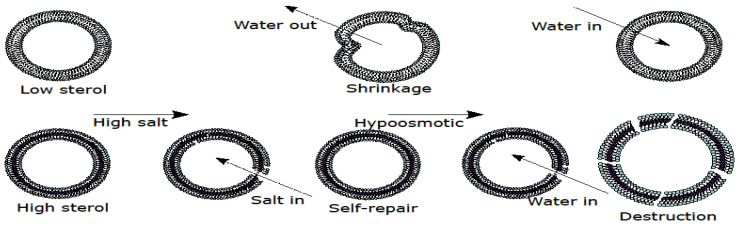
A scheme illustrating how sterol might provoke the pore formation upon changes in the osmotic pressure. High rigidity and low water permeability provided by high sterol might prevent the volume decrease upon high salt stress. In the case the hyperosmotic stress is followed by the hypoosmotic one, sterol-rich structures appear to be prone to rupturing. See text (Introduction) for details.

**Figure 2 membranes-12-01278-f002:**
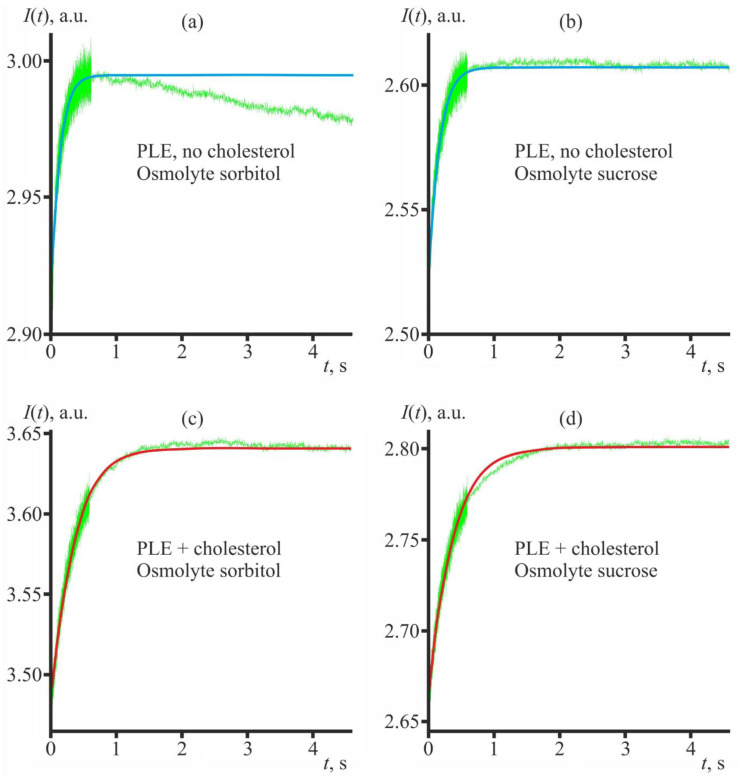
Representative dependences *I*(*t*) of the light intensity scattered from LUV suspensions after application of the hyperosmotic gradient. (**a**,**b**)—LUVs made from PLE (no cholesterol) with sorbitol (**a**) or sucrose (**b**) used as an osmolyte. (**c**,**d**)—LUVs made from PLE + cholesterol lipid mixture with sorbitol (**c**) or sucrose (**d**) used as an osmolyte. Solid blue and red curves represent the fit of the experimentally determined dependence *I*(*t*) (green traces) by Equation (3). The membrane permeability with respect to water determined from the fits is *P_f_* = 5.7 µm/s (PLE + cholesterol membrane) and *P_f_* = 12.7 µm/s (PLE membrane) both for sucrose and sorbitol used as an osmolyte. All curves are rather well fitted by a single-component Lambert function.

**Figure 3 membranes-12-01278-f003:**
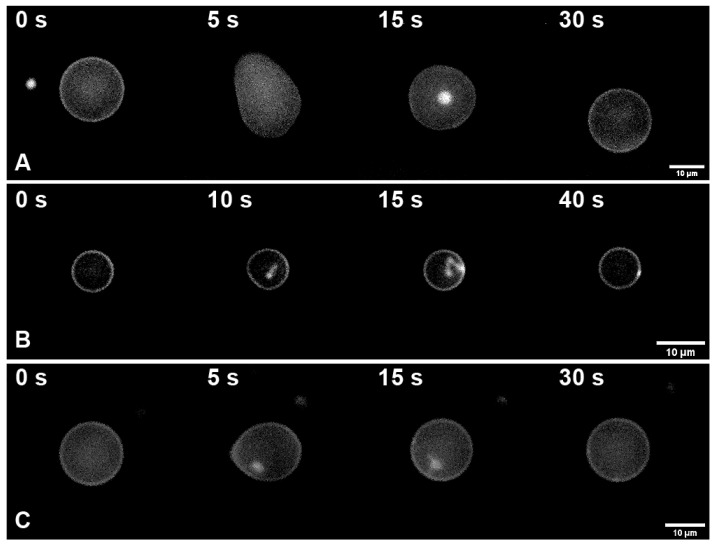
Time course of the GUV reaction on the transient hyperosmotic stress. (**A**)—GUV formed from DOPC; (**B**)—GUV formed from 70 mol.% DOPC + 30 mol.% Ergosterol; (**C**)—GUV formed from 70 mol.% DOPC + 30 mol.% Cholesterol. Time in seconds passed from the application of the hyperosmotic stress is indicated in left upper corners. Time count starts from the moment of the application of the hyperosmotic solution. Scale bar is 10 μm.

**Figure 4 membranes-12-01278-f004:**
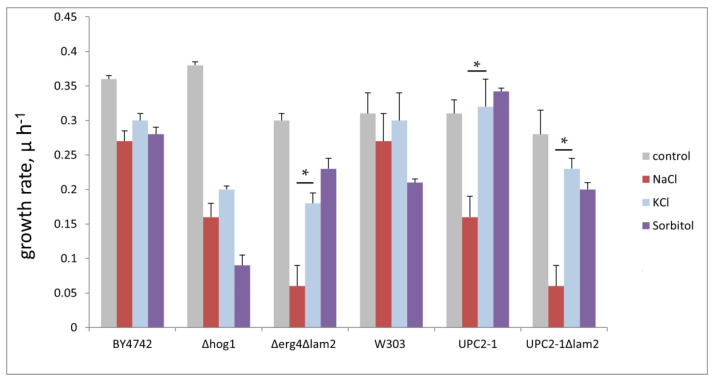
*S. cerevisiae* mutants with elevated PM sterol (*Δ**erg4**Δ**lam2, upc2-1 and upc2-1**Δ**lam2*) are more sensitive to hyperosmotic stress induced by 0.6 M NaCl, but not 0.6 M KCl or 1.2 M sorbitol. *Y*-axis: growth rate, division time equals 0.434/µ. * *p* < 0.05, Two-sample *t*-test for independent samples.

**Figure 5 membranes-12-01278-f005:**
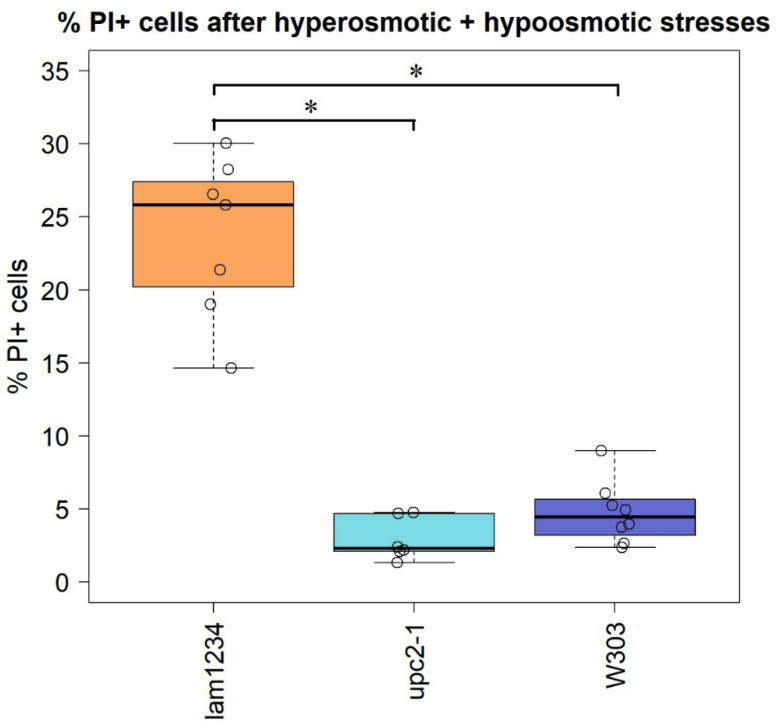
*S. cerevisiae**Δ**lam1**Δ**lam2**Δ**lam3**Δ**lam4* (lam 1234) mutant cells accumulate propidium iodide after hypoosmotic stress. Circles are individual experimental repeat measurements. * *p* < 0.05, Two-sample *t*-test for independent samples.

**Table 1 membranes-12-01278-t001:** Yeast strains used in this study.

Strain	Genotype	Parental Strain	Reference
*W303*	*MATa ade2–101 his3–11 trp1–1 ura3–52 can1–100 leu2–3*	*W303*	Laboratory of A. Hyman
*UPC2-1*	*MATa UPC2–1 ura3–1 his3–11,- 15 leu2–3,-112 trp1–1*	*W303*	[23]
*UPC2-1* *Δ* *lam2*	*MATa UPC2–1 ura3–1 his3–11,- 15 leu2–3,-112 trp1–1* *Δ* *lam2::TRP1*	*W303*	This study
*Δ* *lam1* *Δ* *lam2* *Δ* *lam3* *Δ* *lam4*	*MATa ade2–101 his3–11 trp1–1 ura3–52 can1–100 leu2–3 MATa ade2–101 his3–11 trp1–1 ura3– 52 can1–100 leu2–3* *Δ* *lam3::kanMX4* *Δ* *lam2::TRP1*	*W303*	[23]
*BY4742*	*MATalpha his3* *Δ* *1 leu2* *Δ* *0 met15* *Δ* *0 ura3* *Δ* *0*	*BY4742*	Deletion collection [24]
*Δ* *hog1*	*MATalpha his3* *Δ* *1 leu2* *Δ* *0 met15* *Δ* *0 ura3* *Δ* *0 hog1::kanMX4*	*BY4742*	Deletion collection [24]
*Δ* *erg4* *Δ* *lam2*	*MATalpha his3* *Δ* *1 leu2* *Δ* *0 met15* *Δ* *0 ura3* *Δ* *0 erg4::kanMX4* *Δ* *lam2::HIS3*	*BY4742*	[25]

**Table 2 membranes-12-01278-t002:** Reduction of the average diameter of GUVs depending on the time from the start of hyperosmotic stress. Errors are standard deviations of the corresponding values.

GUV Composition	15 s from the Start of Application of the Hyperosmotic Solution	30 s from the Start of Application of the Hyperosmotic Solution
DOPC	12 ± 5%	3 ± 2%
70 mol.% DOPC + 30 mol./% Ergosterol	5 ± 3%	2 ± 1%
70 mol.% DOPC + 30 mol./% Cholesterol	4 ± 2%	3 ± 2%

## Data Availability

Not applicable.

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
