# Peer review of "Structural Role of Plasma Membrane Sterols in Osmotic Stress Tolerance of Yeast Saccharomyces cerevisiae"

_membranes, 2022, doi:10.3390/membranes12121278_

Round 1
Reviewer 1 Report
It is known that sterols affect the rigidity, fluidity, and permeability of the lipid bilayer of both artificial and biological membranes. The fact that these lipids play an important role in the osmotic processes associated with dehydration and rehydration of cells is also beyond doubt. However, what is the contribution of sterols, the main pool of which is concentrated in plasma membranes, to the adaptation of yeast cells to osmotic/salt stress is not entirely clear.
In addition, literature data are rather contradictory. On the one hand, cells deprived of the usual composition of sterols suffer more in osmotic processes. On the other hand, activation of the HOG pathway for the synthesis of glycerol simultaneously suppresses the synthesis of sterols.
The authors formulated a hypothesis according to which a high content of membrane sterols provokes the formation of pores in the membrane during dehydration and, due to this, there are no significant changes in the cell volume during salt stress. During rehydration, when a hypertonic environment changes to a hypotonic environment, rigid plasma membranes rich in sterols, on the contrary, have a greater tendency to rupture.
Using liposomes (LUV GUV) and yeast strains with different sterol contents, this hypothesis is being tested. Data are presented on the growth of cells with altered content and distribution of sterols between the plasma membrane and their intracellular pool in osmotic processes. In contrast to [4], where it was shown that sterol depletion leads to cell damage both under hyperosmotic exposure and subsequent rehydration, in this work, it was studied how the disruption of sterol gradient and homeostasis affects cell adaptation.
The authors explain the absence of changes in the volume of GUV by balancing of the osmotic gradient during the formation of pores. Could these processes be associated with the presence of membranous protrusions and vesicles inside the cholesterol-rich GUV? (Claessens et al, 2008; Osmotic shrinkage and reswelling of giant vesicles composed DOPG and cholesterols).
244 cholesterol-rich membranes
395 Yeast S. cerevisiae appear to be unique in their strategy of adaptation to the osmotic challenges. They decrease PM sterol content upon hyperosmotic stress. This increases water efflux from the cell upon hyperosmotic stress.
The change in the content of sterols is the result of the activation of the HOG pathway after hyperosmotic stress. The high water membrane permeability cannot increase or decrease the water fluxes; only the time to reach osmotic equilibrium depends on the magnitude of the gradient. Tolerance to dehydration is, first of all, the synthesis of osmolytes.
356 When sterol synthesis is inhibited in Dunalliela, glycerol biosynthesis is disrupted (Zelazny et al, 1995; Plasma Membrane Sterols Are Essential for Sensing Osmotic Changes in the Halotolerant Alga Dunaliella).
Fig. 4 and 5 sign Y-axes.
Author Response
Reviewer 1
- The authors explain the absence of changes in the volume of GUV by balancing of the osmotic gradient during the formation of pores. Could these processes be associated with the presence of membranous protrusions and vesicles inside the cholesterol-rich GUV? (Claessens et al, 2008; Osmotic shrinkage and reswelling of giant vesicles composed DOPG and cholesterols).
Yes, we agree that the formation of protrusions could add to the volume stability of cholesterol-rich GUVs. We modified the text accordingly, and also added the reference.
- The high water membrane permeability cannot increase or decrease the water fluxes; only the time to reach osmotic equilibrium depends on the magnitude of the gradient.
We agree, and we removed the statement from the abstract.
- Tolerance to dehydration is, first of all, the synthesis of osmolytes
We agree that “tolerance to dehydration is, first of all, the synthesis of osmolytes”, and we added this to the text (Discussion). Still, our main point is that sterols increase rigidity of the LUVs, GUVs and the plasma membrane. This additional rigidity prevents swelling upon hyperosmotic stress that, in turn, promotes membrane rupturing upon the following hypoosmotic stress. We modified Introduction to make this point clearer.
- When sterol synthesis is inhibited in Dunalliela, glycerol biosynthesis is disrupted (Zelazny et al, 1995; Plasma Membrane Sterols Are Essential for Sensing Osmotic Changes in the Halotolerant Alga Dunaliella).
We did not see this paper before. It points to an additional link between osmo-tolerance and sterols. We referred to in the Discussion.
- 4 and 5 sign Y-axes.
Done.
Reviewer 2 Report
In their paper, Sokolov et al. study the influence of osmotic stress on the yeast Saccharomyces cerevisiae, in particular focusing on plasma membrane sterol levels. According to their hypothesis, lower sterol content bestows yeast cells with higher resilience towards hyper/hypo-osmotic stress. The manuscript is thought-through, well-written and mostly the messages are clearly stated. Generally, the work is technically sound; my main concern is that I do not see conclusions drawn fully supported by data.
In particular:
11) The authors are interested in the role of sterols in the response of S. cerevisiae to osmotic stress yet they sometimes use E.coli polar lipid extract for creating their vesicle model systems (LUVs). This seems an odd choice, considering that the lipid composition of the Gram-negative bacterium E.coli is very different from the eukaryote S. cerevisiae (phosphatidylglycerols and cardiolipins (negatively charged) as well as lipids prone to form non-lamellar phases (cardiolipins, phosphatidylethanolamines) in E.coli; phosphatidylcholines and -inositols and some phosphatidylethanolamines in S. cerevisiae. It was also not clear to me, when and why they used DOPC (for GUVs apparently) and sometimes PLE. DOPC was also not mentioned in the chemicals section. Can the authors clarify this?
22) The statements of significant differences (e.g. Fig 4, line 308) should be supported by appropriate significance tests.
3) This is actually my main concern. The authors tested 17 strains with defects in ergosterol biosynthesis and/or transport, out of which only 3 showed the behavior they expected. What is their explanation for that? Is the actual effect t of these mutations on the ergosterol levels in the plasma membrane known? Do the authors have any indication that the plasma membrane ergosterol levels are different between the strains that showed different sensitivities and those that did not? Could it actually be other factors and how can this be excluded?
44 ) The authors argues that mutant Δlam1Δlam2Δlam3Δlam4 has elevated plasma membrane ergosterol levels (similar to ∆erg4∆lam2, UPC2-1, UPC2-1∆lam2), but it does not show differential sensitivity towards NaCl, KCl and sorbitol (as the other three do). How do the authors explain this? On the other hand, UPC2-1 did not accumulate PI. It would be interesting to see the PI accumulation in UPC2-1 Δlam2.
Minor comments:
11) In Figure 1, the scenario low cholesterol/water out may be better depicted with a “wobbly” plasma membrane (I mean visualizing the excessive membrane due to decreased cell volume). At the moment, it looks like membrane has disappeared.
22) I got confused about the mutants. Please indicate in Fig 4 which mutants can be expected to have elevated and decreased plasm membrane cholesterol.
In conclusion, based on the shown data, I am not entirely convinced about the author’s conclusions. The GUV experiments are convincing, but the experiments on yeast do not seem to be in full accordance with them. It seems that some mutants showed the expected behavior, and some did not, and the authors focused on the former ones to support their hypothesis.
Author Response
Reviewer 2
- The authors are interested in the role of sterols in the response of cerevisiae to osmotic stress yet they sometimes use E.colipolar lipid extract for creating their vesicle model systems (LUVs). This seems an odd choice, considering that the lipid composition of the Gram-negative bacterium E.coli is very different from the eukaryote S. cerevisiae (phosphatidylglycerols and cardiolipins (negatively charged) as well as lipids prone to form non-lamellar phases (cardiolipins, phosphatidylethanolamines) in E.coli; phosphatidylcholines and -inositols and some phosphatidylethanolamines in S. cerevisiae. It was also not clear to me, when and why they used DOPC (for GUVs apparently) and sometimes PLE. DOPC was also not mentioned in the chemicals section. Can the authors clarify this?
LUVs and GUVs were made of different lipid mixtures to show irrelevance of the vesicle size, the lipid content, and the type of osmolyte on the sterol membrane activity.
- The statements of significant differences (e.g. Fig 4, line 308) should be supported by appropriate significance tests.
Done, for Figs 4 and 5.
- This is actually my main concern. The authors tested 17 strains with defects in ergosterol biosynthesis and/or transport, out of which only 3 showed the behavior they expected. What is their explanation for that? Is the actual effect t of these mutations on the ergosterol levels in the plasma membrane known? Do the authors have any indication that the plasma membrane ergosterol levels are different between the strains that showed different sensitivities and those that did not? Could it actually be other factors and how can this be excluded?
erg mutations lead to the accumulation of precursors of ergosterol in the plasma membrane (PM), lam mutations increase PM ergosterol. Our point is that some of PM ergosterol disruptions caused a relative increase in the sensitivity to NaCl. None of such mutations caused a relative increase in the sensitivities neither to KCl nor to sorbitol. We added this line to the text describing the Results section.
- The authors argues that mutant Δlam1Δlam2Δlam3Δlam4 has elevated plasma membrane ergosterol levels (similar to ∆erg4∆lam2, UPC2-1, UPC2-1∆lam2), but it does not show differential sensitivity towards NaCl, KCl and sorbitol (as the other three do). How do the authors explain this?
Short answer is that we do not know. Possibly, in the quadruple mutant some compensatory mechanisms are activated. Again, our argument is the same as in the case of other erg and lam mutants (answer to the previous point): the mutations either increase the relative sensitivity to NaCl or not. Relative sensitivities to the two other, less aggressive osmolytes, are never increased.
- On the other hand, UPC2-1 did not accumulate PI.
We were also surprised by this result, and we discussed it in the Discussion:
Unexpectedly, the mutant UPC2-1 cells did not accumulate PI (Figure 5), despite elevated PM ergosterol [23]. Possibly, the activity of the reverse transporters in this mutant, Lam proteins, provides local relaxation of the PM rigidity. In other words, despite the total elevated sterol content, even local low-sterol PM patches, might provide sufficient flexibility to allow, as shown in Figure 1, the PM shrinkage upon the hyperosmotic stress, and in this way prevent the influx of the PI-containing external medium, when the osmotic pressure is normalized.
- It would be interesting to see the PI accumulation in UPC2-1 Δlam2.
We agree. Our prediction is that under the conditions we used (Fig. 5), lam2 mutation, as well the quadruple Δlam1Δlam2Δlam3Δlam4 mutation, will lead to an increase in the PI accumulation in the majority of mutant backgrounds. As, for technical reasons, such an experiment is rather laborious, we believe that this is a subject for an independent study.
Minor comments:
- In Figure 1, the scenario low cholesterol/water out may be better depicted with a “wobbly” plasma membrane (I mean visualizing the excessive membrane due to decreased cell volume). At the moment, it looks like membrane has disappeared.
Yes, we agree. We modified the scheme.
- I got confused about the mutants. Please indicate in Fig 4 which mutants can be expected to have elevated and decreased plasm membrane cholesterol.
We modified the figure legend. Now it is:
- cerevisiae mutants with elevated PM sterol (∆erg4∆lam2, upc2-1 and upc2-1∆lam2) are more sensitive to the hyperosmotic stress induced by 0.6M NaCl, but not 0.6M KCl or 1.2M sorbitol.
In conclusion, based on the shown data, I am not entirely convinced about the author’s conclusions. The GUV experiments are convincing, but the experiments on yeast do not seem to be in full accordance with them. It seems that some mutants showed the expected behavior, and some did not, and the authors focused on the former ones to support their hypothesis
“It seems that some mutants showed the expected behavior, and some did not”. Yes, this is essentially the same concern, which was discussed in points 3 and 4. As stated there, our answer is the following: some of PM ergosterol disruptions caused a relative increase in the sensitivity to NaCl. None of such mutations caused a relative increase in the sensitivities neither to KCl nor to sorbitol.
To summarize, it is difficult even to design an experiment to measure the rigidity of the plasma membrane. For this reason, the data on S. cerevisiae, while being in agreement with the hypothesis (Fig. 1), obviously do not prove it. At the same time, the data on LUVs and GUVs provide a more direct argument supporting the hypothesis.
Round 2
Reviewer 2 Report
I support publication as is